# Brassica Extracts Prevent Benzo(a)pyrene-Induced Transformation by Modulating Reactive Oxygen Species and Autophagy

**DOI:** 10.3390/ijms26199519

**Published:** 2025-09-29

**Authors:** José Benito Montes-Alvarado, Paula Garcia-Ibañez, Diego A. Moreno, Fabiola Lilí Sarmiento-Salinas, Xiadani Edén Susano-Hernández, Karen Andrea Larrauri-Rodríguez, Francisco Jesús García-Hernández, Lorena Milflores-Flores, Fabiola Domínguez, Paola Maycotte

**Affiliations:** 1Biotechnology Laboratory, Centro de Investigación Biomédica de Oriente, Instituto Mexicano del Seguro Social, Puebla 74360, Mexico; leon240780@gmail.com; 2Aquaporins Group, Plant Nutrition Department, Centro de Edafología y Biología Aplicada del Segura, Consejo Superior de Investigaciones Científicas, Campus Universitario de Espinardo-25, E-30100 Murcia, Spain; pgibanez@cebas.csic.es; 3Phytochemistry and Healthy Food Lab (LabFAS), Centro de Edafología y Biología Aplicada del Segura, Consejo Superior de Investigaciones Científicas, Campus Universitario de Espinardo-25, E-30100 Murcia, Spain; dmoreno@cebas.csic.es; 4Secretaría de Ciencia, Humanidades, Tecnología e Innovación (SECIHTI), Mexico City 03940, Mexico; lilith_spp@hotmail.com; 5Metabolic Biochemistry Laboratory, Centro de Investigación Biomédica de Oriente, Instituto Mexicano del Seguro Social, Puebla 74360, Mexico; xiadanie.susanohernandez@viep.com.mx (X.E.S.-H.); kareenlarrauri@hotmail.com (K.A.L.-R.); francisco.garcia1303@uppuebla.edu.mx (F.J.G.-H.); 6Facultad de Ciencias Biológicas, Benemérita Universidad Autónoma de Puebla, Puebla 74360, Mexico; lorena.milflores@correo.buap.mx; 7Programa de Ingeniería en Biotecnología, Universidad Politécnica de Puebla, Puebla 72640, Mexico

**Keywords:** Brassicas, broccoli, cabbage, breast cancer, reactive oxygen species, autophagy

## Abstract

Plants from the Brassicaceae family are characterized by their high content of glucosinolates (GSLs), whose hydrolysis products, isothiocyanates (ITC) or indole compounds, have been found to have anti-inflammatory, antioxidant and metabolic regulatory functions. In this work, we used a model of transformation using the MCF10A cell line, a non-tumorigenic breast fibrocystic disease cell line, treated with benzo(a)pyrene (B(a)P), a potent carcinogen known to induce the production of reactive oxygen species (ROS) and DNA damage. Broccoli sprout (BSE) or red cabbage aqueous (RCA) extracts were rich in ITC and indole compounds. Their use decreased B(a)P induced cellular proliferation and ROS production. in addition, RCA extract induced autophagy in MCF10A cells. Our results indicate a potential use of BSE or RCA for the prevention of carcinogen-induced transformation and of RCA as a method for autophagy, a tumor suppressor pathway, induction.

## 1. Introduction

The Brassicaceae family includes plants of nutritional and economical relevance such as broccoli, kale, cabbage, radish and mustard. Glucosinolates (GSLs), a group of sulfur-containing secondary metabolites, are particularly abundant in these plants and are responsible for their peculiar taste [1]. GSLs are enzymatically hydrolyzed by a thioglucosidase, commonly referred to as myrosinase upon tissue damage caused by herbivores, mechanical injury or pathogenesis, cleaving the glucose group from the GSLs. The compounds are then hydrolyzed to thiocyanates, isothiocyanates (ITC), nitriles, epithionitriles or indoles depending on the constitution of the parent glucosinolate molecule, the hydrolysis conditions and the presence of additional specifier proteins [1,2,3]. GSL-hydrolysis products have a variety of biological functions: they may function as a defense mechanism against pathogens, insects and weeds; have been shown to have antibacterial properties; can downregulate Phase I and upregulate Phase II drug metabolism enzyme activities in mammalian cells; and they can regulate inflammation and reactive oxygen species (ROS) production [3,4]. Among the products of GSL hydrolysis, ITCs have been found to have the highest biological activity, with their role as regulators of drug metabolism, hydrogen sulfide production, anti-inflammatory effects, the regulation of the NRF2 (nuclear factor, erythroid 2 like 2) transcription factor and the modulation of autophagy [3] as important mediators for the reported effects on cancer therapy and prevention.

Breast cancer is the most prevalent type of cancer in women worldwide, representing the primary cause of cancer-related mortality among women [5]. Despite advances in targeted therapy, which are most beneficial in early stage patients, advanced-stage breast cancer or patients from the triple-negative (TN) subtype have a poor prognosis and a high recurrence rate [6,7], underscoring the importance of early detection and prevention. Several factors have been associated with an incremented risk of breast cancer and, although genetic factors are important, only 10–15% of breast cancer cases are related to hereditary factors [8]. Other risk factors include being overweight or obese, lack of exercise, alcohol consumption, hormonal therapy, advanced age, having received radiation in the chest area and diverse obstetric and gynecological factors, including early menarche, late menopause and number and timing of childbirth [9,10]. Besides lifestyle changes that decrease exposure to modifiable risk factors, few chemopreventive approaches exist. Those available in a clinical setting include tamoxifen and raloxifene (two selective estrogen receptor modulators) and exemestane (an aromatase inhibitor). These preventive therapies have proved efficient in reducing breast cancer risk, particularly for recurrence prevention, though they have also exhibited adverse secondary effects, such as endometrial cancer, in the case of tamoxifen or raloxifene [11].

Other environmental risk factors have been related to an increased breast cancer risk, including bisphenols, PFAS (per- and polyfluorinated alkyl substances) or exposure to aluminum [10]. The latter’s relationship to breast cancer has been demonstrated in in vitro studies but not in large epidemiological studies. This is partly because of the time needed to develop the disease after an exposure but also because environmental exposure is complex, and likely involves exposure to multiple chemicals at low, continuous doses [10]. Other studies have also suggested a link between red or processed meat consumption and cancer development. Meat is rich in chemical substances such as heterocyclic amines (HCAs), polycyclic aromatic hydrocarbons (PAHs), nitrate and N-nitroso compounds, which have been related to increased risk of multiple cancers [12]. High red meat or processed meat intake has been linked to increased risk of esophageal, endometrial, pancreas, nasopharyngeal carcinoma, lung, gastric, bladder, colorectal, non-Hodgkin lymphoma and breast cancer [12]. Regarding breast cancer, meat consumption has been linked to carcinogenic byproducts formed during high-temperature cooking of red meat, the presence of fat, heme iron and the animal sugar molecule N-glycolylneuraminic acid, which may promote inflammation, oxidative stress and tumor formation. In addition, in some countries, hormonal residues from exogenous hormones used to stimulate the growth of beef cattle can also influence the hormonal load in women [13].

We used a cell transformation model using benzo(a)pyrene (B(a)P), an aromatic hydrocarbon produced during incomplete combustion of meat and other organic materials which is known to induce cell transformation in vitro [14]. B(a)P and its metabolites have been shown to induce DNA adduct formation and DNA strand breaks, as well as to cause ROS production and oxidative damage [14]. Our results show that aqueous broccoli sprout extract (BSE) and red cabbage leaf aqueous (RCA) extracts prevented B(a)P-induced ROS accumulation and proliferation in MCF10A human mammary non-tumorigenic cells. Besides its antioxidant properties, RCA induced autophagy in MCF10A cells, suggesting their potential use as agents for breast cancer prevention.

## 2. Results

### 2.1. Identification and Quantitative Analysis of BSE and RCA Extracts

In this study, we used aqueous broccoli sprout (BSE) or red cabbage leaf (RCA) extracts, both known to be rich in their GSL content. GSL-hydrolysis products have been shown to have important antioxidant, anti-inflammatory, drug detoxicant, chemotherapeutic and cancer preventive roles [3]. We identified the ITC composition (sulforaphane (SFN) or iberin (IB)), as well as the indole 3 carbinol (I3C) content, in both the BSE and RCA extracts. Table 1 shows the ITC and I3C content of both extracts according to the concentrations used in further experiments. SFN was the most abundant ITC in both extracts, and all the analytes were more abundant in the RCA than in the BSE extract if similar concentrations were compared.

### 2.2. Benzo(a)pyrene (B(a)P) Induced ROS Production in Mammary Epithelial Cells

B(a)P is a prototypical environmental pollutant and carcinogen, whose effects on cellular function include DNA damage, ROS production and epigenetic modulation of oncogenes or tumor suppressors, having mutagenic and carcinogenic effects [15]. A 2 μM B(a)P treatment induced an increase in ROS production at 3 h, which later returned to normal levels at 6 h (Figure 1A,B). Similar effects were observed with 4 μM but not with 1 μM B(a)P. Thus, we chose 2 μM B(a)P for the rest of the experiments.

### 2.3. BSE and RCA Extracts Decreased B(a)P-Induced Proliferation and ROS Production

Previous results from our group using the same extracts have shown that doses lower than 2.5 mg/mL of BSE or 50 μg/mL of the RCA extracts do not have an important effect on cellular proliferation of the MCF10A cell line [16]. Accordingly, 150 μg/mL or 1 mg/mL BSE and 10 μg/mL or 25 μg/mL RCA extracts did not induce differences in MTT (Thiazolyl Blue Tetrazolium Bromide) conversion after a 24 h treatment (Figure 2A) or after a 72 h treatment, as measured by cellular confluency (Figure 2B,E). A 24 h 2 μM B(a)P treatment increased MTT conversion in MCF10A cells, suggesting an induction of cellular proliferation. Importantly, all the tested doses of both extracts were able to prevent this increase in MTT conversion (Figure 2C). When cellular proliferation was measured as cellular confluency (Figure 2D,E), despite an apparent increase in confluency observed with the B(a)P treatment, this was not statistically significant at 72 h, indicating that the B(a)P treatment did not increase cellular confluency up to 72 h of treatment and that the observed increase in MTT conversion might be due to changes in cellular metabolism induced by B(a)P and not by increased proliferation at this time point. These results indicate an early metabolic change induced by B(a)P treatment that can be prevented by BSE or RCA extracts. Importantly, no important changes in cell morphology were observed after a 48 h B(a)P treatment with or without the extracts (Figure 2E).

To further evaluate the role of B(a)P on cellular proliferation, we performed a long-term clonogenic assay, where cells were plated and treated with B(a)P± BSE or RCA for 72 h. The medium was replaced with fresh media without treatment and allowed to grow for 8 days. The results are shown on Figure 3. When measured after 8 days of exposure, B(a)P induced proliferation in the MCF10A cells, and this increase in proliferation was prevented by BSE (Figure 3A,B) or RCA (Figure 3A,C) treatment.

Finally, both the BSE and RCA (Figure 4) extracts prevented the increase in ROS induced by a 3 h B(a)P treatment. Altogether, our data indicates an important role for BSE and RCA extracts in the prevention of the characteristics of malignancy induced by B(a)P.

### 2.4. RCA Extracts Induced Autophagy in Mammary Epithelial Cells

Autophagy has been proposed to be an important tumor suppressor pathway, mostly by its effects on the regulation of cellular homeostasis. Since ITCs have been shown to have important chemopreventive functions, we evaluated if this effect was mediated by the promotion of autophagy. We used LC3 processing to evaluate autophagic flux.

During autophagy, LC3I is cleaved and conjugated to phosphatidylethanolamine, forming LC3II, which localizes to the autophagosome and migrates faster in an SDS/PAGE due to its enhanced hydrophobicity. However, when autophagy is induced, the autophagosomal content is degraded after fusion with lysosomes, along with the degradation and recycling of LC3II. Thus, LC3II levels might be increased after autophagy induction (if autophagosomes are increasingly formed) or decreased after autophagy induction (if autophagosomes are being turned over), and these events normally occur sequentially, depending on the treatment exposure time. To evaluate autophagic flux induction, a lysosomal inhibitor is used to induce the accumulation of LC3II [17,18]. For an autophagy inducer, the treatment with the lysosomal inhibitor should increase LC3II levels more than the treatment or the inhibitor alone, reflecting autophagosome induction and degradation in the lysosome, which can be blocked by lysosomal inhibition, as can be observed in starvation-induced autophagy [17,18]. We used chloroquine (CQ) as a lysosomal inhibitor for the last 2 h of a 24 h treatment with the extracts.

To test the preventive effect of the extracts to B(a)P or exposure to other carcinogens, we tested the effect of both BSE and RCA on autophagy without B(a)P treatment. Since we observed similar effects of both doses of the extracts used in the previously tested parameters, we chose a single concentration for both extracts. We found increased LC3II levels in RCA + CQ when compared to control cells and to CQ-only treated cells, indicating autophagy induction by RCA but not BSE and possibly implicating autophagy in the prevention of carcinogenesis with RCA treatment (Figure 5).

B(a)P itself has also been implicated in the induction of endoplasmic reticulum stress and autophagy in different models or doses [19,20]. We tested if B(a)P induced autophagy in the MCF10A cells and the effect of both extracts on this process. We found no changes in LC3II with B(a)P ± CQ treatment suggestive of induced autophagy (Appendix A). B(a)P induced a decrease in p62, and this decrease was prevented by CQ treatment, suggesting autophagy induction. Thus, B(a)P also modulated autophagy in this model, suggesting an important role for this process in B(a)P-induced transformation. The effect of autophagy on B(a)P-mediated carcinogenesis remains to be studied.

## 3. Discussion

B(a)P is a group I carcinogen produced by the combustion of fossil fuels, wood and organic material. It is present in cigarette smoke and in food products processed at high temperatures and has been found in the atmosphere, surface water and soil [21]. Its effects on mutagenesis and carcinogenesis include DNA adduct formation and the production of ROS after being metabolized, leading to DNA damage and mutations, transformation and ultimately carcinogenesis [21,22]. In breast tissue, B(a)P is suggested to accumulate in the fat cells of the breast, where it is known to induce chromosomal aberrations, ROS production and oxidatively induced clustered DNA lesions [21]. It has also been shown to induce proliferation, ROS production and DNA strand breaks in mammary epithelial cells in culture [23], highlighting its determining effect on the promotion of cellular transformation. One of the proposed mechanisms by which B(a)P induces DNA adduct formation suggests that after B(a)P activation by cytochrome P450 (CYP450) to epoxides, these epoxides are converted to diols by epoxide hydrolases and then to a diol epoxide by CYP450. This is a reactive metabolite that can react with DNA to form stable adducts. Alternatively, diols can undergo autoxidation, forming quinones and directly binding DNA or entering a redox cycling pathway, forming ROS and leading to DNA damage. ROS amplification would then induce DNA damage, mutations, cell transformation and carcinogenesis [22].

Cancer prevention strategies involve early detection and decreasing the exposure to cancer risk factors, such as obesity or smoking, among others. In the context of hereditary breast and/or ovarian cancer syndromes, primary prevention includes health counselling, environmental controls, prophylactic surgery and chemoprevention based on absolute cancer risk, family history and patient’s preferences [24]. Thus, chemoprevention, or the use of a synthetic, natural or biological agent at a safe dose to reduce the risk of developing cancer has emerged as a promising strategy to lower cancer risk, particularly for those people with a higher risk of cancer development [24]. Few chemopreventive options exist and, for breast and ovarian cancer, the options include raloxifene, tamoxifen and combined oral contraceptive use, with important side effects limiting their use in clinical practice. In this work, we describe the effective use of broccoli sprout (BSE) and red cabbage aqueous (RCA) extracts in decreasing cancer related features induced by B(a)P, including cellular metabolic alterations measured by MTT conversion (Figure 2), cellular proliferation (Figure 3) and ROS production (Figure 4).

Increased proliferation induced by B(a)P has been proposed to be due to its carcinogenic effects. As mentioned previously, its metabolic activation induces DNA adducts, strand breaks, mutations, chromosomal aberrations, ROS production, oxidatively induced DNA lesions and tumorigenesis. Moreover, B(a)P has been shown to induce cell cycle alterations (decrease in G1 and increase in G2/M and S phases), indicative of increased proliferation [22,23]. Our data shows changes in MTT conversion at 24 h induced by B(a)P (Figure 2C) with no changes in cell confluency until 72 h (Figure 2B,D) and no evident effect on cellular morphology until this time point (Figure 2E). However, when the 72 h treatment was removed and the cells were allowed to grow after 8 days, an increase in cell confluency was induced by B(a)P treatment (Figure 3). Since the MTT assay measures dehydrogenase activity [25], these results indicate that B(a)P induces an early metabolic shift in MCF10A cells, which increases their long-term proliferation, possibly inducing metabolic alterations and/or ROS production.

Oxidative stress or increased ROS production are common cellular events occurring after the exposure to many cancer risk factors including tobacco, particulate matter, infectious agents, obesity, carcinogen exposure and aging [26]. Thus, antioxidant use has been suggested as a potential cancer preventive strategy. In this regard, GSLs and their derivatives are almost exclusive and abundant in plants from the Brassicacea family, which includes broccoli, cabbage, red radish, Brussels sprouts and mustards [27]. GSL derivatives, including ITC and indole derivatives have been shown to have important effects on cellular function [3]. Among them, SFN is the most studied ITC, and among its best characterized functions is the activation of NRF2, a transcription factor regulating multiple cytoprotective responses, including the antioxidant response, drug metabolism, protein homeostasis, immune response, lipid, carbohydrate or iron metabolism and autophagy [3]. SFN, as well as other ITCs, has also been shown to regulate the activity of CYP450, which is responsible for Phase I xenobiotic detoxification, as well as carcinogen activation [3]. Although we did not measure NRF2 or CYP450 activation, their regulation of antioxidant gene expression or drug metabolism could be responsible for the observed effects. B(a)P is known to activate the aryl-hydrocarbon receptor (AHR), which activates CYP450 and ROS production [15]. CYP450 activation is also responsible for the formation of B(a)P metabolites responsible for its mutagenic and carcinogenic effects. Thus, CYP450 inhibition, as well as NRF2 activation by the ITCs present in the extracts, could be responsible for the prevention of ROS production and the decrease in proliferation observed with the extracts in MCF10A-B(a)P-treated cells.

Autophagy has been shown to regulate cellular homeostasis by removing damaged (ROS-producing) mitochondria, by degrading oncogenic viruses, by maintaining the cellular genomic stability and by inducing oncogene-induced senescence [28]. Thus, autophagy is a homeostatic mechanism with an important role on cellular and organismal maintenance, as well as on lifespan extension [29]. RCA extracts induced autophagy in MCF10A cells, as measured by LC3 processing, possibly implicating autophagy induction in the prevention of carcinogenesis induced by B(a)P and indicating an important role for RCA extracts in the promotion of autophagy for the prevention of important diseases. Since both extracts contain ITCs, but we could only observe autophagy induction with the RCA extract, it is unlikely that ITCs are implicated in autophagy-induction. One of the most important differences between both extracts is the high phenolic, and particularly anthocyanin, content of RCA [30]. Since anthocyanins have been shown to promote autophagy in breast cancer cell lines [31], it will be important to evaluate the effect of anthocyanins on autophagy induction in future studies.

Since both broccoli sprouts and red cabbage are rich sources of ITCs, with reported SFN concentrations of 1153 mg /100 g in the case of broccoli sprouts [32] or 0.758 mg/100 g in the case of red cabbage [33], and the latter are also a rich source of phenolic compounds, particularly anthocyanins [30], the reported effects are likely achievable through diet. Thus, a diet rich in plants from the Brassicaceae family is likely to prevent carcinogenesis by their antioxidant properties and autophagy-inducing potential in the case of red cabbage. Although other mechanisms are likely to be involved in the prevention of carcinogenesis induced by Brassica extracts, possibly involving NRF2, CYP450 or the anti-inflammatory effect induced by ITCs, and these specific effects remain to be fully characterized, our data suggests that the general effect will result in the prevention of carcinogenesis, with important consequences for carcinogen-mediated transformation.

## 4. Materials and Methods

### 4.1. Broccoli Sprout Extract (BSE) and GSL-Hydrolysis Product Determination

BSE was prepared as previously described [16,34]. Briefly, broccoli seeds (*Brassica oleracea* var. *italica*) (Intersemillas, S.A., Valencia, Spain) were germinated upon initial imbibition and aeration for 24 h in distilled water with 5 g/L sodium hypochlorite for disinfection. Seeds were extended on inert cellulose (CN SEEDS Ltd., Pymoor, UK) and transferred to a culture chamber. They were kept in the darkness for 3 days and then transferred to light/darkness cycles until day 8. After this time, the sprouts were collected, weighed and frozen in liquid nitrogen. Then, they were lyophilized and ground for extraction. An amount of 50 mg of lyophilized powder was extracted with 1.5 mL of H_2_O, frozen and lyophilized to produce BSE. An aliquot of this extract was incubated for 24 h at 25 °C under agitation to hydrolyze GSLs to their derivatives. After centrifugation and filtration, the sample was analyzed by UHPLC-QqQ-MS/MS (Agilent Technologies, Waldbron, Germany) to identify and quantify ITC and GSL-hydrolysis products according to previously established protocols [35].

### 4.2. Aqueous Red Cabbage (Brassica oleracea L. var capitata f. rubra) Extract (RCA) and GSL-Hydrolysis Product Determination

Red cabbage leaf extracts were prepared as described previously [16,36,37]. Briefly, red cabbage (*Brassica oleracea* L. var. *capitata f. rubra*) seeds from Sakata Seed Iberica (Valencia, Spain) were pre-treated with deionized water and continuous aeration for 24 h, and they were planted in vermiculite for 2 days in darkness at 28 °C and 60% relative humidity. The plants were grown from September 2018 to February 2019 under a semiarid Mediterranean climate and were drip-irrigated with ¼ Hoagland solution. Leaf samples were collected 15 days after the appearance of the flower bud, freeze-dried and ground. The powder was stored at −80 °C until further experiments. Extracts were produced by maceration at 1:15 (*w*:*v*) with water for 4 h in the darkness under continuous agitation. Samples were centrifuged at 13,500× *g* for 15 min. Supernatants were collected and percolated through a filter, frozen and lyophilized to produce RCA. After hydrolysis, the presence of ITCs in the extract was analyzed and quantified using an UHPLC coupled with a 6460-triple quadrupole-MS/MS (Agilent Technologies, Waldbronn, Germany) and a Zorbax Eclipse Plus C18 column (2.1 × 50 mm, 1.8 μm). Sulforaphane (SFN), iberin (IB) and indole-3-carbinol (I3C) from Santa Cruz Biotechnology (Dallas, TX, USA) were used as standards.

### 4.3. Cell Culture and Cell Viability Evaluation

The MCF10A mammary epithelial cell line, originally isolated from a fibrocystic disease in a woman, was originally acquired from the University of Colorado Tissue Culture Core. The cell line was verified by STRs to match MCF10A CRL-10317 ATCC cells in 2023. After thawing an early-passage vial, cells were used for no longer than six months. Cells were cultured in the following media: DMEM/F12 (Caisson Labs Inc., Smithfield, UT, USA) DFP18-1LT, 5% horse serum, 0.5 μg/mL hydrocortisone, 20 ng/mL EGF, 100 ng/mL cholera toxin, 10 μg/mL insulin. B(a)P (Cat. 48564, Supelco, Merck KGaA, Darmstadt, Germany) was resuspended in DMSO and used at the indicated concentrations. Where indicated, DMSO was used as a vehicle control at 0.1%, equivalent to the amount of DMSO in B(a)P treatment.

### 4.4. Cell Viability

Cellular confluence was used to assess cellular viability using an Incucyte SX1 real-time microscope (Sartorius, Göttingen, Germany), and cell confluence was calculated using the Basic Analyzer and AI cell confluency segmentation module of Incucyte SX1 software (2023A Rev2). Pictures were taken every 4 h, and cellular confluency was evaluated with each treatment for 72 h. Cell morphology was assessed visually up to 72 h after treatment. Cell viability was also evaluated using MTT (M5655-1G, Sigma-Aldrich, Merck KGaA, Darmstadt, Germany) reagent. After a 24 h treatment with the extracts ± B(a)P, cells were incubated with 3 mM MTT solution, incubated for 1 h, media was removed and MTT crystals were dissolved with DMSO (472301, Sigma-Aldrich, Merck KGaA, Darmstadt, Germany). Absorbance was read at 570 nm in a Synergy (Biotek, Agilent, Santa Clara, CA, USA) spectrophotometer. Unless otherwise stated, reagents were acquired from Sigma Aldrich Co., St. Louis, MO, USA.

### 4.5. Clonogenic Assay

Long-term effects on cell viability were evaluated using a cellular clonogenic assay, where cells were treated for 72 h with B(a)P ± extracts and washed. The medium was replaced with regular medium without treatment and were allowed to grow for 8 days, replacing the medium with fresh medium at day 3. Afterward, cells were fixed (10% acetic acid, 10% methanol, 80% distilled water) and stained with crystal violet solution (0.4% crystal violet, 20% ethanol). Pictures were taken from all the wells, and the stain was dissolved in 30% acetic acid solution. Absorbance was quantified on a Synergy (Biotek, Agilent, Santa Clara, CA, USA) spectrophotometer at 540 nm.

### 4.6. Reactive Oxygen Species (ROS) Determination

For ROS determination, cells were stained with 10 μM dihydroethidium (DHE, 37291) for 30 min at 37 °C, protected from light. Cells were washed, trypsinized and centrifuged. The pellet was resuspended in PBS with 3% fetal bovine serum (FBS), filtered and analyzed in a BD Facs Canto II flow cytometer (BD Biosciences, Franklin Lakes, NJ, USA), using the PerCP-A channel (Ex: 488 nm, Em: 675 nm). Data was analyzed using Flow Jo V 10.0 software. For ROS-high or ROS-low populations, cells were analyzed as described in [38]. We defined 90% of the population in the controls as ROS^high^ and 10% as ROS^low^, and the same gates were used for each treatment.

### 4.7. Western Blot

For autophagy determination, cells were treated with the indicated extracts for 24 h ± 20 μM chloroquine (CQ) for the last 2 h of treatment. We used a 2 h CQ treatment to avoid saturation of autophagosome accumulation by a longer treatment and to decrease cytotoxicity induced by CQ. Cells were lysed in RIPA buffer with protease inhibitors (Complete, Roche, 11697498001, Sigma-Aldrich, Merck KGaA, Darmstadt, Germany). Proteins were quantified with Bradford reagent, and 20 mg were loaded on a 15% SDS-PAGE and transferred to a PVDF membrane. Membranes were blocked with 5% skim milk and probed with an anti-LC3B (NB100-2220, Novus Biologicals, Centennial, CO, USA), anti-p62 (88588, Cell Signaling Technologies, Danver, MA, USA) or anti-β actin (A5441, Sigma-Aldrich, Merck KGaA, Darmstadt, Germany) antibodies overnight at 4 °C. Membranes were later incubated with secondary HRP-linked anti-mouse IgG (A2304, Sigma-Aldrich, Merck KGaA, Darmstadt, Germany) or anti-rabbit IgG (7074 Cell Signaling Technologies, Danver, MA, USA) for 1 h at room temperature. Membranes were developed with HRP-kit (Immobilon Western (WBKLS0500, Millipore, Merck KGaA, Darmstadt, Germany) and scanned on a C-DiGit (LI-COR, Licorbio, Lincoln, NE, USA) scanner. Densitometric analysis was performed using Image Studio 6.0 (LI-COR Biosciences, LLC, Lincoln, NE, USA) software 6.0.0.28 version.

### 4.8. Statistical Analysis

For the statistical analysis, a one-way or two-way ANOVA was used as indicated, followed by Bonferroni or Dunnett’s post hoc tests. Graphs were generated, and statistical analyses were performed using GraphPad Prism 6.

## Figures and Tables

**Figure 1 ijms-26-09519-f001:**
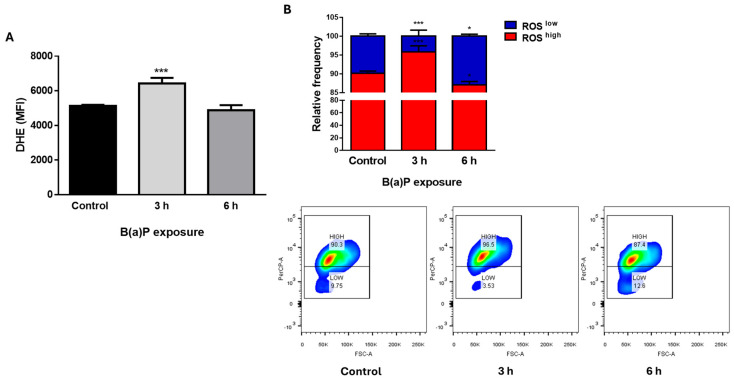
B(a)P treatment induced an increase in cellular ROS levels. MCF10A cells were treated with 2 μM B(a)P and ROS levels were evaluated by dihydroethidium (DHE) staining. B(a)P treatment induced an increase in ROS levels after a 3 h exposure, represented both as the mean fluorescence intensity (MFI, **A**) and as ROS-low and ROS-high populations (**B**). Graphs show mean fluorescence intensity or relative frequencies ± SD of 4–5 independent experiments. One-way (**A**) or two-way ANOVA (**B**), with Dunnett’s post hoc tests. * difference with respect to the control, * *p* < 0.05; *** *p* < 0.001.

**Figure 2 ijms-26-09519-f002:**
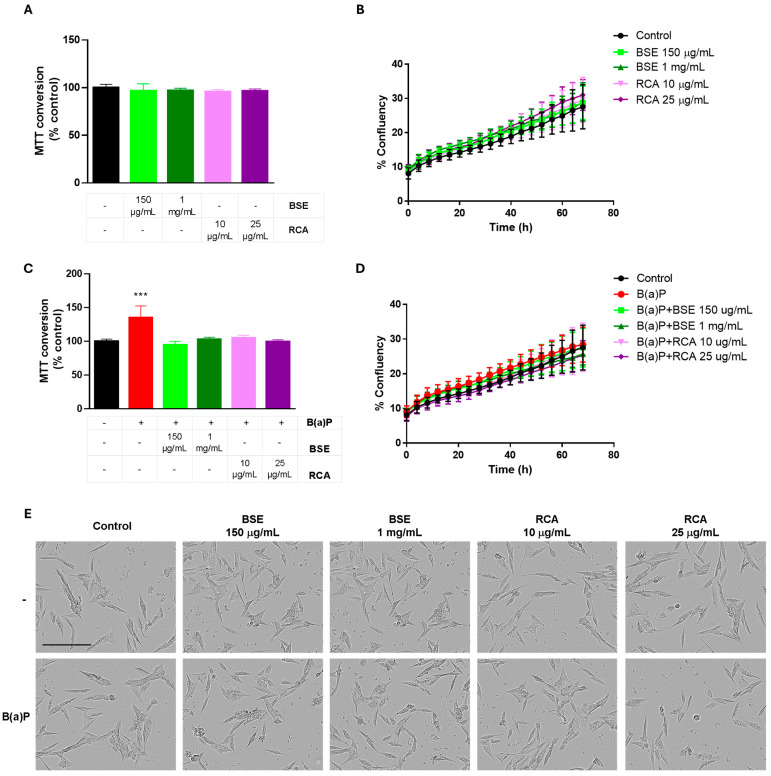
Broccoli sprout extract (BSE) or red cabbage aqueous extract (RCA) decreased B(a)P-induced metabolic changes in MCF10A cells. In (**A**,**B**), MCF10A cells were treated with BSE or RCA at the indicated concentrations, and cellular viability was evaluated by MTT conversion (**A**) or by cellular confluency evaluated by time-lapse microscopy (**B**). A 2 μM B(a)P treatment for 24 h induced an increase in MTT conversion, which was prevented by BSE or RCA treatment (**C**). Cellular confluency was not affected by B(a)P with or without the extracts after 72 h (**D**). No changes in cell morphology were observed with B(a)P with or without the extracts after a 48 h treatment (**E**). Graphs show mean MTT conversion or confluency ± SD of three independent experiments. One-way ANOVA, with Dunnett’s post hoc tests was used. * shows the difference with respect to the control, *** *p* < 0.001. The bar in (**E**) represents 200 µm.

**Figure 3 ijms-26-09519-f003:**
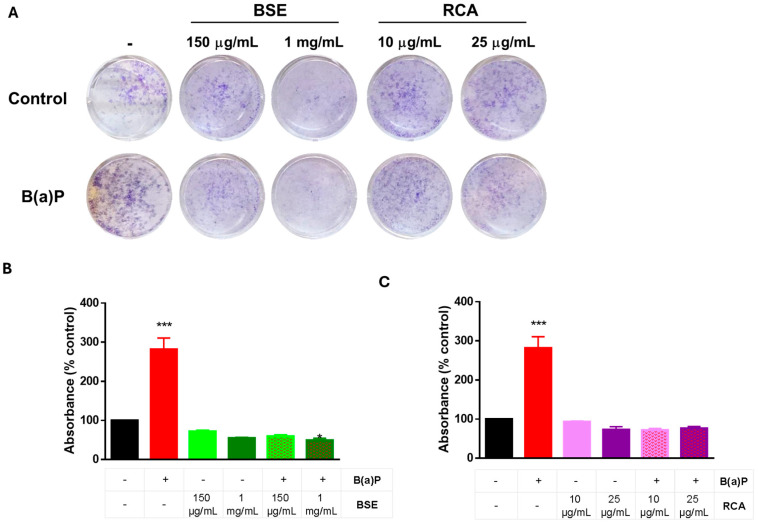
B(a)P induced cell proliferation in a long-term clonogenic assay, and this effect was prevented with the BSE or RCA extracts. Cells were treated with 2 μM B(a)P, along with BSE (**A**,**B**) or RCA (**A**,**C**) extracts at the indicated concentrations for 72 h; the media was replaced with fresh medium without treatment, and the cells were allowed to proliferate for 8 days, then fixed and stained. Long-term clonogenic assays show that both BSE and RCA at both concentrations tested were able to prevent B(a)P-induced proliferation in MCF10A cells. Graphs show mean normalized crystal violet absorbance ± SD of 3–5 independent experiments. One-way ANOVA with Dunnett’s post hoc was test was used. * shows difference with respect to the control, *** *p* < 0.001.

**Figure 4 ijms-26-09519-f004:**
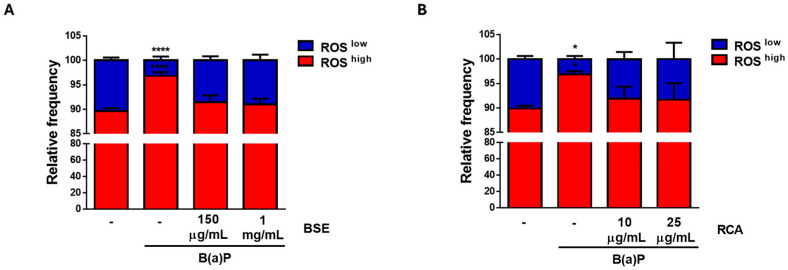
BSE and RCA extracts prevented B(a)P-induced ROS production. MCF10A cells were treated with B(a)P along with BSE (**A**) or RCA (**B**) at the indicated concentrations for 3 h, and ROS were evaluated with DHE staining and flow cytometry. The graphs show mean relative frequencies ± SD of 3–5 independent experiments. Two-way ANOVA with Dunnett’s post hoc test; * vs. control, * *p* < 0.05; **** *p* < 0.0001.

**Figure 5 ijms-26-09519-f005:**
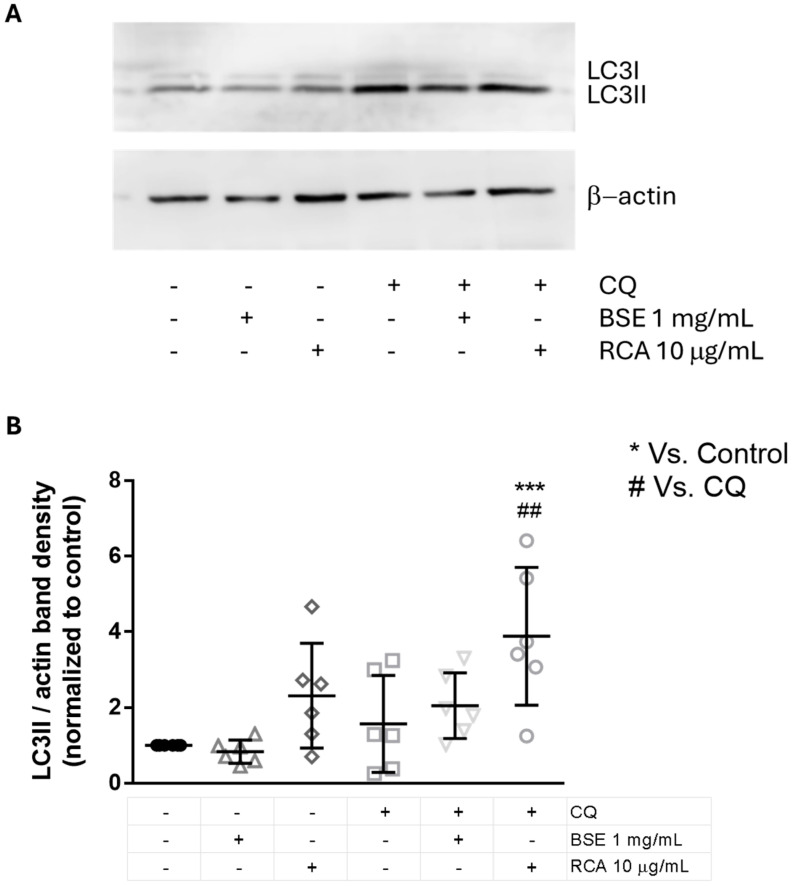
The RCA extract induced autophagy in a breast immortalized cell line. MCF10A cells were treated with BSE or RCA at the indicated concentrations for 24 h with or without 20 µM chloroquine (CQ) for the last 2 h of treatment (**A**). The graph in (**B**) shows the mean normalized band density ± SD of the band density of LC3II normalized to β-actin from 6 independent experiments. One-way ANOVA, Bonferroni post hoc test. ## *p* < 0.01 compared to CQ alone; *** *p* < 0.001 compared to the control.

**Table 1 ijms-26-09519-t001:** Isothiocyanate (ITC) and I3C concentrations in each of the extracts.

**Source**	**Type of Extract**	**Metabolite**	**150 μg/mL**	**1 mg/mL**
Broccoli sprouts (*Brassica oleracea* var. *italica)*	Aqueous (BSE)	SFN	804 nM	5360 nM
		IB	23.9 nM	159.2 nM
		I3C	13.25 nM	88.3 nM
**Source**	**Type of Extract**	**Metabolite**	**10 μg/mL**	**25 μg/mL**
Red cabbage (*Brassica oleracea* L. var. *capitata f. rubra*) leaf extracts	Aqueous (RCA)	SFN	210 nM	525 nM
		IB	39.9 nM	99.8 nM
		I3C	112.5 nM	281.3 nM

SFN, sulforaphane; IB, iberin; I3C, indole 3 carbinol.

## Data Availability

The original contributions presented in this study are included in the article/Appendix A. Further inquiries can be directed to the corresponding authors.

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
