# Peer review of "Brassica Extracts Prevent Benzo(a)pyrene-Induced Transformation by Modulating Reactive Oxygen Species and Autophagy"

_ijms, 2025, doi:10.3390/ijms26199519_

Round 1
Reviewer 1 Report
Comments and Suggestions for Authors
The aim of the reviewed manuscript was to demonstrate the possibility of using plant extracts from broccoli sprouts and red cabbage in preventing carcinogen-induced transformation by reducing ROS levels and inducing autophagy.
Below are some comments on the manuscript.
- Results
2.1. Benzo(a)pyrene (B(a)P) induces proliferation and ROS production in mammary epithelial cells.
Description 2.1. suggests that benzo(a)pyrene induces proliferation, however, the manuscript lacks results and their discussion confirming this. In this part of the results, only the ROS level was discussed, which may indicate oxidative stress or mitochondrial damage.
Figure 1. I propose to standardize the captions in the figures. The abbreviation C should be replaced with the caption Control. Please remove the abbreviation ns from the chart.
2.2. I suggest that the order of presentation of the results be reversed. The results from 2.2 should be presented first, followed by those from 2.1.
In this part of the manuscript, representative UHPLC-QqQ-MS/MS chromatograms from the analyzed samples should be shown.
2.3. In Figure 2 A and C, I propose replacing the term MTT transformation with conversion. In general, the MTT test determines % cell viability.
Representative time-lapse microscopy images are missing in Figure 2B and D. Significance is missing in the graphs showing percent confluence.
2.4. Comment on line 161/162.
Autophagy is a process in which autophagic vacuoles (autophagosomes) fuse with primary lysosomes, resulting in digestion. Autophagosomes are not degraded in lysosomes. The only digestion that can occur in lysosomes is the so-called microautophagy.
In the manuscript, the authors write that autophagy has an effect on preventing carcinogenesis induced by B(a)P in the case of RCA treatment.
It should be explained why the experiment determining the LC3 level did not show the effect of benzo(a)pyrene and its combined effect with the extracts tested. The literature shows that benzo(a)pyrene itself also affects the induction of autophagy. It should be checked how plant extracts modulate the level of autophagy after combined action with benzo(a)pyrene.
Please also provide information on why chloroquine was used for the last 2 hours of the 24-hour treatment with extracts. Most often, preincubation of cells with chloroquine is used several hours before the administration of the tested compound.
Why were only single concentrations used for BSE and RCA in the assessment of LC3 levels?
The authors write: “We found increased levels of LC3II in RCA+CQ compared to control cells and cells treated with CQ alone, which indicates the induction of autophagy by RCA but not BSE and probably implicates autophagy in the prevention of B(a)P-induced carcinogenesis in RCA treatment”.
The combination of RCA with chloroquine does not indicate the induction of autophagic processes, but rather the accumulation of autophagic vacuoles in the cytoplasm. According to the mechanism of action, chloroquine blocks the fusion of autophagosomes and lysosomes, so autophagy does not occur. In this way, stress and apoptosis are most often induced in cells.
- Results
4.3. Please provide information on the percentage of DMSO in the control and what the MTT dye was dissolved in.
Is 1 h of incubation sufficient to reduce the MTT dye?. The standard incubation time for dye reduction in MTT methods is 2 to 4 h.
There is no full information on how cell confluence was determined. Was a cell segmentation algorithm used?
Discussion
The discussion should include an explanation of why only the RCA extract induced autophagy. The authors are trying to explain the mechanism of action of the tested extracts on a molecular basis, but their considerations are not supported by any results.
Summary, the manuscript requires thorough modification. It should be expanded with additional analyses. In order to demonstrate the cytotoxic effect of benzo(a)pyrene and the protective effect of the tested extracts, I propose to additionally perform an analysis of the level of apoptosis (annexin V test). Proliferation in the cells studied can be further extended to include a scratch wound healing test and cell cycle. There is no photographic documentation. All results should also be standardized, i.e. factor/dose/time. In its current form, the manuscript is not suitable for publication in IJMS due to the very basic results and requires supplementation with more detailed data.
Author Response
The aim of the reviewed manuscript was to demonstrate the possibility of using plant extracts from broccoli sprouts and red cabbage in preventing carcinogen-induced transformation by reducing ROS levels and inducing autophagy.
Below are some comments on the manuscript.
- Results
2.1. Benzo(a)pyrene (B(a)P) induces proliferation and ROS production in mammary epithelial cells.
Comment 1 Description 2.1. suggests that benzo(a)pyrene induces proliferation, however, the manuscript lacks results and their discussion confirming this. In this part of the results, only the ROS level was discussed, which may indicate oxidative stress or mitochondrial damage.
Response to Comment 1Thank you for your comment. This was our mistake. Former section 2.1 (now section 2.2) should only state: Benzo(a)pyrene (B(a)P) induced ROS production in mammary epithelial cells. This has been corrected in the new section 2.2. We agree with the reviewer that proliferation was not properly addressed or discussed. A long-term cell confluency test was incorporated on Figure 2E-G. These results demonstrate that B(a)P induced cell proliferation which was prevented by both extracts. Also, a discussion on both the results and discussion sections were incorporated to explain these results.
Comment 2 Figure 1. I propose to standardize the captions in the figures. The abbreviation C should be replaced with the caption Control. Please remove the abbreviation ns from the chart.
Response to Comment 2. Thank you for your comment. The captions in the figures were standardized and all Cs were replaced with the caption Control. Abbreviation ns was removed from all the charts.
Comment 3. 2.2. I suggest that the order of presentation of the results be reversed. The results from 2.2 should be presented first, followed by those from 2.1.
Response to Comment 3. Thank you for your comment. The order was reversed and the results from the previous section 2.2 are now on section 2.1 and viceversa.
Comment 4. In this part of the manuscript, representative UHPLC-QqQ-MS/MS chromatograms from the analyzed samples should be shown.
Response to Comment 4. Thanks for your request. Since the equipment used for the sulforaphane, iberin and indole-3-carbinol analysis did not include a DAD detector, we do not have a complete chromatogram for both extracts. Chromatograms for specific analyte quantification based on transitions (specific and documented fragmentation of these compounds) are shown on our response letter (attached). Specifically, they correspond to: A) Sulforaphane from BSE, B) iberin from BSE, C) indole-3-carbinol from BSE, D) Sulforaphane from RCA, E) iberin from RCA and F) indole-3-carbinol from RCA. As these chromatograms do not add further relevant insights beyond the targeted quantification already presented, we believe their inclusion in the main manuscript is not warranted.
Comment 5. 2.3. In Figure 2 A and C, I propose replacing the term MTT transformation with conversion. In general, the MTT test determines % cell viability.
Response to Comment 5. Thank you for your comment. The labels on the graph were changed to MTT conversion.
Comment 6. Representative time-lapse microscopy images are missing in Figure 2B and D. Significance is missing in the graphs showing percent confluence.
Response to Comment 6. Thank you for your comment. We have included representative time-lapse microscopy images of Figure 2B and D on Supplementary Figure 1. Significance is missing because there were no statistical differences among treatments at the same time points.
2.4. Comment on line 161/162.
Comment 7. Autophagy is a process in which autophagic vacuoles (autophagosomes) fuse with primary lysosomes, resulting in digestion. Autophagosomes are not degraded in lysosomes. The only digestion that can occur in lysosomes is the so-called microautophagy.
Response to comment 7. Thank you for your comment, we apologize for the mistake. We have changed the sentence as follows to avoid confusion as it reads on lines 205-207:
…when autophagy is induced, the autophagosomal content is degraded after fusion with lysosomes, also degrading and recycling LC3II.
Comment 8. In the manuscript, the authors write that autophagy has an effect on preventing carcinogenesis induced by B(a)P in the case of RCA treatment.
It should be explained why the experiment determining the LC3 level did not show the effect of benzo(a)pyrene and its combined effect with the extracts tested. The literature shows that benzo(a)pyrene itself also affects the induction of autophagy. It should be checked how plant extracts modulate the level of autophagy after combined action with benzo(a)pyrene.
Response to comment 8. Thank you for your comment. Originally, we did not include B(a)P treatment because we wanted to evaluate the effect of the extracts before carcinogen exposure to test its preventive effects. We clarify this on section 2.4 as follows:
To test the preventive effect of the extracts to B(a)P or carcinogen exposure, we tested the effect of both BSE and RCA on autophagy without B(a)P treatment. Lines 213-214.
In response to the reviewer´s comment, we tested the combined effect with the extracts, and the results are shown on Supplementary Figure 2 and discussed on section 2.4 (lines 226-233) as follows:
B(a)P itself has also been implicated in the induction of endoplasmic reticulum stress and autophagy in different models or doses [17,18]. We tested if B(a)P induced autophagy in the MCF10A cells and the effect of both extracts on this process. We found no changes in LC3II with B(a)P+/- CQ treatment suggestive of induced autophagy (Supplementary Figure 2). B(a)P induced a decrease in p62, and this decrease was prevented with CQ treatment, suggesting autophagy induction. Thus, B(a)P also modulated autophagy in this model, suggesting an important role of this process in B(a)P-induced transformation. The effect of autophagy on B(a)P- mediated carcinogenesis remains to be studied.
Comment 9. Please also provide information on why chloroquine was used for the last 2 hours of the 24-hour treatment with extracts. Most often, preincubation of cells with chloroquine is used several hours before the administration of the tested compound.
Response to Comment 9. Thank you for your comment. Long-term CQ exposure can be cytotoxic to some cell lines. Besides, we have observed, using the same cell line that this time point is enough to induce an increase in LC3II and it is not saturating. Longer CQ treatments usually increase LC3II to such levels that an increase induced by the treatment is not visible. Since we wanted to test the induction of autophagic flux, we chose the 24-hour time point since we considered that autophagy induction would occur after the increase in ROS (3h). We used CQ for the las 2 hours of treatment to allow autophagosome accumulation induced by each treatment in those 2 hours. This was stated on the Methods section lines 380-382:
We used a 2h CQ treatment to avoid saturation of autophagosome accumulation by a longer treatment and to decrease cytotoxicity induced by CQ.
Comment 10. Why were only single concentrations used for BSE and RCA in the assessment of LC3 levels?
Response to Comment 10. Since similar effects were observed with both doses of BSE and RCA in all the parameters tested, we chose a single concentration for the WB so that we could include all the samples in the same membrane. This was stated on lines 214-216 as follows:
Since we observed similar effects of both doses of the extracts used in the previously tested parameters, we chose a single concentration for both extracts.
Comment 11. The authors write: “We found increased levels of LC3II in RCA+CQ compared to control cells and cells treated with CQ alone, which indicates the induction of autophagy by RCA but not BSE and probably implicates autophagy in the prevention of B(a)P-induced carcinogenesis in RCA treatment”.
The combination of RCA with chloroquine does not indicate the induction of autophagic processes, but rather the accumulation of autophagic vacuoles in the cytoplasm. According to the mechanism of action, chloroquine blocks the fusion of autophagosomes and lysosomes, so autophagy does not occur. In this way, stress and apoptosis are most often induced in cells.
Response to Comment 11. Thank you for your comment. We agree with the author that the combination of RCA with CQ reflects accumulation of autophagic vacuoles. As we clarify on section 2.4, it is the difference between treatment+CQ Vs. CQ alone that reflects autophagic flux induction, since LC3II can be degraded after fusion of the autophagosome with the lysosome. Thus, LC3II levels alone do not reflect autophagy induction/ inhibition. We are basing this analysis on the following references:
- Mizushima, N.; Yoshimori, T. How to interpret LC3 immunoblotting. Autophagy 2007, 3, 542-545, doi:10.4161/auto.4600.
- Klionsky, D.J.; Abdel-Aziz, A.K.; Abdelfatah, S.; Abdellatif, M.; Abdoli, A.; Abel, S.; Abeliovich, H.; Abildgaard, M.H.; Abudu, Y.P.; Acevedo-Arozena, A.; et al. Guidelines for the use and interpretation of assays for monitoring autophagy (4th edition)(1). Autophagy 2021, 17, 1-382, doi:10.1080/15548627.2020.1797280.
And we improved our explanation of section 2.4 (lines 204-215) as follows:
During autophagy, LC3I is cleaved and conjugated to phosphatidylethanolamine, forming LC3II which localizes to the autophagosome and migrates faster in an SDS/PAGE due to its enhanced hydrophobicity. However, when autophagy is induced, the autophagosomal content is degraded after fusion with lysosomes, also degrading and recycling LC3II. Thus, LC3II levels might be increased after autophagy induction (if autophagosomes are increasingly formed) or decreased also after autophagy induction (if autophagosomes are being turned over) and these events normally occur sequentially, depending on the treatment exposure time. To evaluate autophagic flux induction, a lysosomal inhibitor is used to induce the accumulation of LC3II [1,2]. For an autophagy inducer, the treatment with the lysosomal inhibitor should increase LC3II levels more than the treatment or the inhibitor alone, reflecting autophagosome induction and degradation in the lysosome which can be blocked by lysosomal inhibition, as can be observed in starvation-induced autophagy [1,2].
- Results
Comment 12. 4.3. Please provide information on the percentage of DMSO in the control and what the MTT dye was dissolved in.
Response to Comment 12. DMSO was used at 0.1%, equivalent to the amount of DMSO in B(a)P treatment. This was incorporated in the cell culture and cell viability evaluation section as follows:
Where indicated, DMSO was used as a vehicle control at 0.1%, equivalent to the amount of DMSO in B(a)P treatment. Lines 338-339.
Comment 13. Regarding the dissolution of MTT crystals, they were dissolved in DMSO, as specified on the cell culture and cell viability evaluation section:
Response to Comment 13. …cells were incubated with 3mM MTT solution, incubated for 1 h, media was removed and MTT crystals were dissolved with DMSO (Sigma 472301). Lines 343-344.
Comment 14. Is 1 h of incubation sufficient to reduce the MTT dye?. The standard incubation time for dye reduction in MTT methods is 2 to 4 h.
Response to Comment 14. Thank you for your comment. We have used 1 h of incubation with this MTT concentration, since higher incubation times have induced signal saturation and inability to distinguish differences between most wells.
Comment 15. There is no full information on how cell confluence was determined. Was a cell segmentation algorithm used?
Response to Comment 15. Cell confluency was determined using the Incucyte SX1 software. This was specified on the cell culture and cell viability evaluation section as follows:
…cell confluence was calculated using the Basic analyzer and AI cell confluency seg-mentation analysis of the Incucyte SX1 software. Lines 340-341.
Discussion
Comment 16. The discussion should include an explanation of why only the RCA extract induced autophagy.
Response to Comment 16. Thank you for your comment. We included in the discussion the following explanation:
Since both extracts contain ITC but we could only observe autophagy induction with the RCA extract, it is unlikely that ITC are implicated in autophagy-induction. One of the most important differences between both extracts is the high phenolic and anthocyanin content of RCA [3]. Since anthocyanins have been shown to promote autophagy in breast cancer cell lines [4], it will be important to evaluate the effect of anthocyanins on autophagy induction in future studies.
Comment 17. The authors are trying to explain the mechanism of action of the tested extracts on a molecular basis, but their considerations are not supported by any results.
Response to Comment 17. Thank you for your comment. We consider we have shown an effect of both our extracts on the inhibition of cellular proliferation and ROS production. We agree that this is not a full molecular characterization of their effects, but our results provide valuable insights onto their mechanism of action. Since our data shows an important effect of both extracts on B(a)P induced transformation, we consider our results to be of great importance, since they suggest that an increase in brassica consumption could prevent carcinogen-mediated transformation.
Comment 18. Summary, the manuscript requires thorough modification. It should be expanded with additional analyses. In order to demonstrate the cytotoxic effect of benzo(a)pyrene and the protective effect of the tested extracts, I propose to additionally perform an analysis of the level of apoptosis (annexin V test).
Response to Comment 18. Thank you for your comment. B(a)P doses used in this study are not intended to be cytotoxic but rather induce cellular transformation and increase proliferation. At the doses used, we did not see any cytotoxic effect (Figure 2 and Supplementary Figure 1). Thus, an analysis of apoptosis/ cell death induction is not warranted.
Comment 19. Proliferation in the cells studied can be further extended to include a scratch wound healing test and cell cycle. There is no photographic documentation.
Response to Comment 19. Thank you for your comment. We performed a long-term clonogenic assay to further test the effect of B(a)P on the induction of proliferation and the preventive effect of both extracts. In this experiment, cells were treated for 72 h (similar to the ones shown on Fig. 2B and D), the treatment was removed and replaced with fresh media and cells were allowed to grow for 8 days. Results are shown on Figure 2E, a long-term increase in cell number was observed with B(a)P treatment which could be prevented with both extracts. Also, photographic documentation for the data on Figures 2B and D is included in Supplementary Figure 1.
Comment 20. All results should also be standardized, i.e. factor/dose/time. In its current form, the manuscript is not suitable for publication in IJMS due to the very basic results and requires supplementation with more detailed data.
Response to Comment 20. Thank you for your comment. We performed extract dose and time standardization on a previous study (Sanchez-Guzman et al, J Med Food, 2024) as mentioned on section 2.3. B(a)P concentration was also standardized as mentioned on section 2.2, based on their ROS-inducing capability we chose the 2 mM concentration, as mentioned on lines 119-122:
2 mM B(a)P treatment induced an increase in ROS production at 3h which later returned to normal levels at 6 h (Figure 1A, B). Similar effects were observed with 4 mM but not with 1 mM B(a)P (data not shown). Thus, we chose 2 mM B(a)P for the rest of the experiments.

Reviewer 2 Report
Comments and Suggestions for Authors
The authors studied the chemopreventive activity of aqueous extracts from broccoli sprouts (BSE) and red cabbage (RCA) in countering benzo(a)pyrene (B(a)P)-induced transformation in MCF10A breast epithelial cells. The results show that both extracts mitigate B(a)P-induced metabolic changes and reactive oxygen species (ROS) production, while RCA additionally promotes autophagy. The findings support that Brassica-derived compounds have a great potential in cancer prevention. The research is interesting. Before the manuscript is suggested for publication in the journal, some concerns need to be addressed.
Major concerns:
- Please discuss the physiological relevance of the concentrations of BSE and RCA used in this study. Are such concentrations (e.g., 150 ug/mL and 1 mg/mL BSE; 10 ug/ mL and 25 ug/ mL) achievable through diet?
- Figure 2 shows that BSE and RCA extracts reduce B(a)P-induced cell proliferation using 24-h MTT assay and 72-h confluency assay. To demonstrate that the Brassica extracts prevent B(a)P-induced transformation, long term transformation assay (e.g., colony formation or soft agar assay) is suggested. Including DNA damage assays (e.g., comet assay and γ-H2AX staining) would strengthen conclusions about transformation suppression.
- The authors claim that RCA extract induces autophagy in MCF10A cells by assessing LC3-II levels (Figure 3). It is good to provide additional autophagy markers such as p62 degradation or autophagosome formation (via imaging or electron microscopy) to demonstrate autophagic flux.
Minor concerns:
- References were inconsistently formatted.
Author Response
The authors studied the chemopreventive activity of aqueous extracts from broccoli sprouts (BSE) and red cabbage (RCA) in countering benzo(a)pyrene (B(a)P)-induced transformation in MCF10A breast epithelial cells. The results show that both extracts mitigate B(a)P-induced metabolic changes and reactive oxygen species (ROS) production, while RCA additionally promotes autophagy. The findings support that Brassica-derived compounds have a great potential in cancer prevention. The research is interesting. Before the manuscript is suggested for publication in the journal, some concerns need to be addressed.
Major concerns:
Comment 1. Please discuss the physiological relevance of the concentrations of BSE and RCA used in this study. Are such concentrations (e.g., 150 ug/mL and 1 mg/mL BSE; 10 ug/ mL and 25 ug/ mL) achievable through diet?
Response to Comment 1. Thank you for your comment. We discuss this relevance in the Discussion section (lines 323-328) as follows:
Since both broccoli sprouts and red cabbage are rich sources of ITC with reported SFN concentrations of 1153mg /100 g in the case of broccoli sprouts [1] or 0.758 mg/100 g in the case of red cabbage [2]; and the latter being also a rich source of phenolic compounds particularly anthocyanins [3], the reported effects are likely achievable through diet. Thus, a diet rich in plants from the Brassicaceae family is likely to prevent carcinogenesis by their antioxidant properties and autophagy-inducing potential in the case of red cabbage.
Comment 2. Figure 2 shows that BSE and RCA extracts reduce B(a)P-induced cell proliferation using 24-h MTT assay and 72-h confluency assay. To demonstrate that the Brassica extracts prevent B(a)P-induced transformation, long term transformation assay (e.g., colony formation or soft agar assay) is suggested. Including DNA damage assays (e.g., comet assay and γ-H2AX staining) would strengthen conclusions about transformation suppression.
Response to Comment 2. Thank you for your comment. We included a long term clonogenic assay (Figure 2, F, G). In this experiment (72 hour treatment, medium replaced with normal media and allowed to grow for 8 days), B(a)P induced cellular proliferation, its effects were prevented by BSE or RCA treatment strengthening our results on the prevention of cellular proliferation. We also did a g-H2AX WB, we found increased g-H2AX with BaP treatment but our extracts did not decrease its levels after BaP treatment. We did not include this result on our figures because we did not have enough repetitions for it to be included and did not have enough time to perform a time-course in order to evaluate different time points and test whether the extracts could decrease g-H2AX at different time points. However, we will test the mechanism by which both extracts prevent proliferation and ROS induction by BaP and if this effect is independent of g-H2AX in future studies.
Comment 3. The authors claim that RCA extract induces autophagy in MCF10A cells by assessing LC3-II levels (Figure 3). It is good to provide additional autophagy markers such as p62 degradation or autophagosome formation (via imaging or electron microscopy) to demonstrate autophagic flux.
Response to Comment 3. Thank you for your comment. We initially did not include p62 because its expression is regulated by both ROS and NRF2 [4]. Since B(a)P induces ROS and ITC induce NRF2 activation, we considered this would be a confounding factor. We measured autophagic flux by using a lysosomal inhibitor (CQ) to block LC3II degradation during autophagosome degradation after fusion with the lysosome. As we clarify on section 2.4, the difference between treatment+CQ Vs. CQ alone reflects autophagic flux induction, since CQ would block LC3II degradation with treatment Vs. without treatment and is a measurement of autophagosome production with the treatment. We are basing this analysis on the following references:
- Mizushima, N.; Yoshimori, T. How to interpret LC3 immunoblotting. Autophagy 2007, 3, 542-545, doi:10.4161/auto.4600.
- Klionsky, D.J.; Abdel-Aziz, A.K.; Abdelfatah, S.; Abdellatif, M.; Abdoli, A.; Abel, S.; Abeliovich, H.; Abildgaard, M.H.; Abudu, Y.P.; Acevedo-Arozena, A.; et al. Guidelines for the use and interpretation of assays for monitoring autophagy (4th edition)(1). Autophagy 2021, 17, 1-382, doi:10.1080/15548627.2020.1797280.
And improved our explanation of section 2.4 (lines 204-215) as follows:
During autophagy, LC3I is cleaved and conjugated to phosphatidylethanolamine, forming LC3II which localizes to the autophagosome and migrates faster in an SDS/PAGE due to its enhanced hydrophobicity. However, when autophagy is induced, the autophagosomal content is degraded after fusion with lysosomes, also degrading and recycling LC3II. Thus, LC3II levels might be increased after autophagy induction (if autophagosomes are increasingly formed) or decreased also after autophagy induction (if autophagosomes are being turned over) and these events normally occur sequentially, depending on the treatment exposure time. To evaluate autophagic flux induction, a lysosomal inhibitor is used to induce the accumulation of LC3II [5,6]. For an autophagy inducer, the treatment with the lysosomal inhibitor should increase LC3II levels more than the treatment or the inhibitor alone, reflecting autophagosome induction and degradation in the lysosome which can be blocked by lysosomal inhibition, as can be observed in starvation-induced autophagy [5,6].
Nevertheless, to answer the reviewer, we include a p62 WB on Supplementary Figure 2. In this WB, p62 decreased after extract treatment suggesting increased autophagy, but its degradation was not prevented by CQ treatment, making the interpretation confusing. Since p62 can activate NRF2 through Keap1 binding and NRF2 can regulate p62 expression, these parameters could interfere/ compensate with p62 degradation by autophagy.
Minor concerns:
Comment 4. References were inconsistently formatted.
Response to Comment 4. Thank you for your comment. We have revised and corrected the reference format.

Round 2
Reviewer 1 Report
Comments and Suggestions for Authors
Below are comments regarding the revised manuscript.
- I believe that the results from the Supplementary Materials should be incorporated into the manuscript. In their current form, the Figures are modest and could be expanded with additional results from the analyses performed, especially since the confluence results are included in Figure 1B and the photographic documentation is in the Supplementary Materials.
- In Figure 2E, the images of the clonogenic assay are difficult to read. Well-formed colonies are not visible on the plates. The quality of the images should be improved.
Note regarding the clonogenic assay:
In the classic clonogenic assay, colonies after crystal violet staining are counted manually or using software, not based on absorbance measurement. The test result is the number of colonies or the fraction of cells surviving after exposure to the agent (the number of colonies formed compared to the control).
In modified versions of the assay, absorbance is sometimes measured after dissolving crystal violet, which allows for a semi-quantitative assessment of total colony biomass but does not distinguish between individual colonies and is less precise in the classic clonogenic survival analysis.
Because the authors have photographic documentation, I suggest supplementing the results with the determination of a cell survival curve, as is performed in the standard assay (the protocol can be used: https://doi.org/10.1038/nprot.2006.339).
The description under Figure 2 F and G should also be updated. The graphs represent the mean (of what?) ± SD from 3–5 independent experiments.
- I propose splitting subsection 4.3 into the following: Cell culture, cell viability, and clonogenic assay. Line 357-diH2O - spelling should be corrected.
- Section 4 - Materials and Methods is missing the methodology for analyzing morphological changes in cells. There is also no separate discussion in the Results section.
In summary, the authors have responded to the reviewer's comments and improved the manuscript. It is now time to address the submitted comments and modify the manuscript.
Author Response
Reviewer 1
Comment 1. I believe that the results from the Supplementary Materials should be incorporated into the manuscript. In their current form, the Figures are modest and could be expanded with additional results from the analyses performed, especially since the confluence results are included in Figure 1B and the photographic documentation is in the Supplementary Materials.
Response 1. Thank you for your comment. The images from Supplementary Figure 1 are now included in Figure 1E.
Comment 2. In Figure 2E, the images of the clonogenic assay are difficult to read. Well-formed colonies are not visible on the plates. The quality of the images should be improved.
Response 2. Thank you for your comment. We have changed the pictures on previous Figure 2E, now Figure 3A. The quality of the images has been improved. However, MCF10A cells do not form tight colonies and rather grow in a diffuse pattern, as shown on the pictures on Figure 2E. So, well-formed colonies are not formed and only diffuse colonies can be seen on the plate.
Comment 3. Note regarding the clonogenic assay:
In the classic clonogenic assay, colonies after crystal violet staining are counted manually or using software, not based on absorbance measurement. The test result is the number of colonies or the fraction of cells surviving after exposure to the agent (the number of colonies formed compared to the control).
In modified versions of the assay, absorbance is sometimes measured after dissolving crystal violet, which allows for a semi-quantitative assessment of total colony biomass but does not distinguish between individual colonies and is less precise in the classic clonogenic survival analysis.
Because the authors have photographic documentation, I suggest supplementing the results with the determination of a cell survival curve, as is performed in the standard assay (the protocol can be used: https://doi.org/10.1038/nprot.2006.339).
Response 3. Thank you for your comment. We followed the protocol mentioned above to perform our clonogenic assays. However, as mentioned previously, MCF10A do not form tight colonies and only diffuse colonies can be seen on the plate. Thus, we consider that dye extraction and absorbance measurements are more precise than colony counting, since colonies cannot be easily distinguished as shown on Figure 3A.
Comment 4. The description under Figure 2 F and G should also be updated. The graphs represent the mean (of what?) ± SD from 3–5 independent experiments.
Response 4. Thank you for your comment. All the figure legends were revised to more clearly describe the graph measurements shown.
Comment 5. I propose splitting subsection 4.3 into the following: Cell culture, cell viability, and clonogenic assay. Line 357-diH2O - spelling should be corrected.
Response 5. Thank you for your comment. Section 4.3 has been split as suggested. Also, diH2O was changed to distilled water.
Comment 6. Section 4 - Materials and Methods is missing the methodology for analyzing morphological changes in cells. There is also no separate discussion in the Results section.
Response 6. Thank you for your comment. Section 4.4, line 390 now reads:
Cell morphology was assessed visually up to 72 h after treatment.
Also, in the discussion, we added the following sentence on lines 286-288:
Our data shows changes in MTT conversion at 24 h induced by B(a)P (Figure 2C) with no changes in cell confluency until 72 h (Figure 2B,D) and no evident effect on cellular morphology until this time point (Figure 2E).
In summary, the authors have responded to the reviewer's comments and improved the manuscript. It is now time to address the submitted comments and modify the manuscript.

Reviewer 2 Report
Comments and Suggestions for Authors
The manuscript has been well revised.
Author Response
Thank you for your comments. We appreciate your input.
Round 3
Reviewer 1 Report
Comments and Suggestions for Authors
The authors addressed the reviewer's comments and revised the manuscript.